# Interplay between the Endogenous Opioid System and Proteasome Complex: Beyond Signaling

**DOI:** 10.3390/ijms20061441

**Published:** 2019-03-21

**Authors:** Francesca Felicia Caputi, Laura Rullo, Serena Stamatakos, Sanzio Candeletti, Patrizia Romualdi

**Affiliations:** Department of Pharmacy and Biotechnology, *Alma Mater Studiorum*—University of Bologna, Irnerio 48, 40126 Bologna, Italy; francesca.caputi3@unibo.it (F.F.C.); laura.rullo3@unibo.it (L.R.); serena.stamatakos2@unibo.it (S.S.); sanzio.candeletti@unibo.it (S.C.)

**Keywords:** opioid system, opioid receptors, GPCRs, 26S proteasome, ubiquitination, UPS

## Abstract

Intracellular signaling mechanisms underlying the opioid system regulation of nociception, neurotransmitters release, stress responses, depression, and the modulation of reward circuitry have been investigated from different points of view. The presence of the ubiquitin proteasome system (UPS) in the synaptic terminations suggest a potential role of ubiquitin-dependent mechanisms in the control of the membrane occupancy by G protein-coupled receptors (GPCRs), including those belonging to the opioid family. In this review, we focused our attention on the role played by the ubiquitination processes and by UPS in the modulation of opioid receptor signaling and in pathological conditions involving the endogenous opioid system. The collective evidence here reported highlights the potential usefulness of proteasome inhibitors in neuropathic pain, addictive behavior, and analgesia since these molecules can reduce pain behavioral signs, heroin self-administration, and the development of morphine analgesic tolerance. Moreover, the complex mechanisms involved in the effects induced by opioid agonists binding to their receptors include the ubiquitination process as a post-translational modification which plays a relevant role in receptor trafficking and degradation. Hence, UPS modulation may offer novel opportunities to control the balance between therapeutic versus adverse effects evoked by opioid receptor activation, thus, representing a promising druggable target.

## 1. Introduction

The endogenous opioid system comprises four major families of opioid ligands: β-endorphins, enkephalins, dynorphins, and nociceptin/orphanin FQ [1,2,3]. These opioid neuropeptides and their cognate receptors are widely distributed across the neuraxis, and, in particular, in pain pathways [1,2,4,5]. In addition to pain modulation, they also participate in the control of several different functions including addiction, stress responses, depression, anxiety, gastrointestinal transit, and the neuroendocrine and immune functions [6,7,8]. Following agonist activation, either endogenous or exogenous, the inhibitory G proteins (G_αi_–G_αo_) dissociate and subsequently engage a variety of effectors that basically depress neural functions [4,9] through the inhibition of adenylate cyclase (AC) and ion channels modulation [10,11,12,13]. The signaling modulation in the synaptic transmission carried out by opioid receptors is crucial at pre- and post-synaptic levels. Indeed, their activation triggers a cascade of events causing the reduction of neurotransmitter release and membrane hyperpolarization [14,15]. Beyond the G-protein mediated signal, agonist binding to the opioid receptors may also cause the recruitment of different arrestin effectors, affecting the balance between the different effects (therapeutic and adverse) evoked by opioid receptor activation [16,17,18,19,20].

The presence of the proteasome complex and ubiquitin (Ubq) protein in the synaptic termination, beyond their existence at cytosolic and nuclear levels, suggests a crucial role played by the ubiquitin-proteasome system (UPS) in the regulation of synapse molecular content. Indeed, UPS contribution to the functional reorganization of synapses [21,22,23] could affect receptor signaling and neuronal functioning [24,25,26]. For instance, ubiquitin-dependent mechanisms may control the membrane occupancy of many receptors, thus, regulating the G protein-coupled receptor (GPCR) endocytosis or internalization signaling, and also the GPCRs receptor level itself [27,28,29]. In this regard, the opioid receptor family is composed of different GPCRs [µ-opioid receptor (MOR), δ-opioid receptor (DOR), κ-opioid receptor (KOR), and nociceptin (NOP) opioid receptors] that transduce the physiological signal of endogenously produced neuropeptides as well as trigger the effect of exogenously administered opiate drugs. Since these receptors can regulate a broad range of effectors, the knowledge of molecular mechanisms regulating their intracellular signaling appears crucial. In this view, the involvement of ubiquitination processes and UPS complex in the endocytic trafficking (e.g., internalization) and in the down-regulation phenomenon induced by opiate agonists exposure [30,31,32], could represent valuable support in understanding opioid related effects.

## 2. Ubiquitination Processes and Ubiquitin-Proteasome System

The intracellular protein levels depend on the balance between synthesis and degradation processes, both essential for accurate cell functioning. In particular, protein degradation is finely regulated through two main mechanisms: lysosomal digestion [33,34] and degradation through the ubiquitin-proteasome system (UPS) [35,36,37,38]. In particular, the 26S proteasome is a dynamic and extremely abundant protein complex [37,39] endowed with the ability to degrade different intracellular proteins (about 90% of the entire non-lysosomal degradation) as long as they are conjugated with Ubq. Thus, Ubq represents the “label marker” allowing a highly specific proteolysis to prevent uncontrolled protein degradation [39,40,41]. In addition to the degradation of mutated or damaged proteins, UPS participates in the regulation of several cellular processes, such as cell growth and proliferation, cell cycle control through the proteolysis of specific regulatory proteins (e.g., cyclins) [42], DNA repair, and regulation of the immune and inflammatory responses [43,44]. The conjugation of selected substrates with Ubq molecules occurs through the action of three different enzymes: E1 (ubiquitin activating enzyme), E2 (ubiquitin conjugating enzyme), and E3 (ubiquitin ligase enzyme) which work sequentially to label proteins for different fates [45,46,47,48] (Figure 1). 

Notably, the E1 enzyme, thanks to ATP molecule hydrolysis, forms a high-energy tio-esther bond with Ubq involving the thiol group of the E1 enzyme active site. Thus, through a trans-estherification reaction, the activated portion of Ubq is transferred to a cysteine residue on the conjugating enzyme E2. This latter allows the binding of the activated Ubq to the protein substrate which is specifically bound to the enzyme E3 [45] (Figure 1, Ubiquitn conjugation pathway). E3 represents the most important enzymatic class in the whole ubiquitination process as it guarantees the selective recognition of the substrate, ensuring the efficiency of the entire process.

Essentially, proteins can be modified by the conjugation of a single Ubq molecule to one or several lysine residues, thus, resulting in mono- or poly-ubiquitinated products. However, because Ubq itself contains lysine residues that act as sites of self-conjugation, poly-ubiquitin chains can also be subsequently produced [49]. The ubiquitin post-translational modifications, both mono- or poly-ubiquitination, direct the conjugated substrates to different cellular fates.

For instance, the mono-ubiquitination process is particularly involved in the histone regulation [50,51] and also in endocytosis, thus, regulating the activity of several proteins located at the plasma membrane [52,53] (Figure 2).

The most studied poly-ubiquitin chain is linked to the lysine-48 residue, and it is known as a “protein destroyer” because it labels proteins for the 26S proteasomal degradation [54,55]. The poly-ubiquitin chain linked to the lysine-63 is instead degraded via lysosomal pathway [56], and it is also involved in DNA repair [57]; the chain linked to the lysine-11 appear to be directly implicated in cell cycle regulation even though its function is not entirely clear [58,59]. Finally, the lysine-6 poly-ubiquitination seems to be associated with DNA repair [55,59,60] and also with mitochondrial homeostasis [61,62] (Figure 2). 

Independent of the residue on which it takes place, the ubiquitination process is reversible. Indeed, the de-ubiquitination enzymes (DUBs) act to hydrolyze individual linkages to cleave Ubq chains from their substrates [63,64]. For instance, regarding the 26S proteasomal degradation, the ubiquitin chains must be removed from the substrates before translocation within the catalytic core, so ensuring the correctness of protein degradation and allowing Ubq endogenous recycling (Figure 1, Ubiquitin recycling).

The constitutive proteasomal machinery is represented by the 26S proteasome which comprises a proteolytic core (20S core particle, 20S-CP) and two regulatory particles (19S regulatory particles, 19S-RP) and it is highly conserved in eukaryotes [65,66,67] (Figure 3), even though the existence of different and alternative proteasome complexes have been demonstrated [68,69,70]. In this view, several studies indeed described the existence of an immunoproteasome primarily expressed in cells of hematopoietic origin, or in non-hematopoietic cells exposed to inflammatory cytokines [71,72]. In the 26S proteasome, the three proteolytic sites are located in β1, β2, and β5 subunits of the 20S-CP [46], whereas the LMP2 (or β1i), MECL-1 (or β2i), and LMP7 (or β5i) subunits, which derive from a different subset of genes, are the correspondent proteolytically active sites of the immunoproteasome [73,74]. 

When a substrate is bound by a poly-ubiquitin chain linked to the lysine-48 residue, it is marked for the 26S proteasome degradation process. Therefore, after recognition, de-ubiquitination and unfolding processes, the substrate enters into the 20S-CP proteolytic chamber where it is fragmented by the six proteolytic sites (β1, β2, and β5 for each of the two β rings) into peptides ranging from two to 25 amino acid residues [45,46]. Unlike traditional proteases, the 26S proteasome generally shortens several times the smaller protein fragments produced, thus, preventing the accumulation of partially digested proteins inside the cell. As mentioned, the 26S proteasome is composed of a barrel-shaped 20S-CP and two 19S-RP. The 19S-RP is made of different subunit categories, such as Rpn-family located in the LID section, which is involved in the recognition step and in the Ubq-chain removal from the substrates, and the Rpt-family subunits which are instead located in the BASE section and attend to the unfolded proteins translocation into the central chamber. The 20S-CP comprises four heterohepatmeric rings (7 β subunits forming the two inner rings and 7 α subunits forming the outer rings). The internal surface of each β1, β2, and β5 subunit contains the proteolytic site having caspase, trypsin, and chymotrypsin-like activities, respectively. The α rings form an axial pore allowing substrate migration toward the proteolytic chamber. The movement through this channel of only unfolded polypeptides is prevented or permitted by the closing or opening state of the gate formed by the N-terminal tails of the α subunits [46], even though the exact mechanism that modulates the two different gate conformations is not yet completely clear.

## 3. Involvement of UPS Machinery in the Opioid Receptor Signaling Associated with Analgesia, Neuropathic Pain, and Addictive Behavior 

Some evidence indicates that the time-dependent agonist-induced MOR and DOR down-regulation can be attenuated by proteasome inhibitors, and to a lesser extent by lysosomal inhibitors [75]. It is worth noting that in the absence of agonist ligands proteasome inhibitors are able to increase MOR and DOR levels, thus, suggesting a prominent role of UPS either in basal and in agonist-induced turnover of opioid receptors [75] (Table 1).

In addition to these findings, it has also been reported that prolonged morphine exposure promotes the G_β_ down-regulation, an effect that is totally suppressed by MG-115 or lactacystin proteasome inhibitors [76]. Notably, authors suggested that the proteasomal degradation of G_β_ protein participates in the so-called “hypertrophy of the cAMP system” caused by the prolonged morphine-induced MOR activation. 

Based on this evidence, in our laboratory, we compared the effects evoked by the exposure to different opioid agonists on the 26S proteasome activity. Our study revealed interesting preliminary results highlighting that morphine, fentanyl, buprenorphine, and tapentadol, exhibiting different affinity profile for the opioid receptors, produced different alteration at proteasomal level [77]. First, we confirmed the ability of morphine to increase the 26S proteasomal activity after prolonged exposure [78], and we also reported a similar and more rapid effect for fentanyl. The same analysis after buprenorphine and tapentadol showed a different situation since buprenorphine reduced proteasome activity after prolonged exposure whereas tapentadol did not cause significant alterations over time [77] (Table 1). This picture suggested that the specific ubiquitin post-translational modifications, driving proteasomal degradation, may occur differently after exposure to different opioid analgesic drugs. It is likely, and somehow attractive, to postulate that the ubiquitination process evoked by the above-mentioned drugs could promote a different degree of MOR receptor ubiquitination. In addition, it is also conceivable that opioid drugs might promote the increase of other poly-ubiquitinated products, different from opioid receptors. This situation could explain the greater proteolytic activity recorded after prolonged strong opioid agonist exposure.

In this regard, morphine use seems strongly related to oxidative stress and the production of free radicals at the cellular level [79,80]. These phenomena, together with the increase of pro-inflammatory cytokines and chemokine receptors induced by the same drug, appear all related to the loss of opioid analgesia [81,82,83,84] and could evoke an increase of UPS activity [85].

Other studies have shown that overnight exposure to [D-Ala^2^, *N*-MePhe^4^, Gly-ol]-enkephalin (DAMGO) or [D-Pen2,D-Pen5]encephalin (DPDPE), selective MOR and DOR agonists respectively, produces a significant decrease of regulator G protein signaling protein 4 (RGS4) which acts as GTPase that modulates opioid receptor signaling, and causes a profound loss of opioid receptors in SH-SY5Y cells [86]. The RGS4 down-regulation appears completely blocked by MG-132 pretreatment or by the specific proteasome inhibitor lactacystin and, accordingly, the protein remains strongly poly-ubiquitinated. In other words, authors suggest that DAMGO or DPDPE treatment promote the RGS4 poly-ubiquitination which normally acts as a signal degradation for UPS.

In contrast, the opioid receptor loss was not counteracted by MG-132 [86], suggesting that differently from RGS4, MOR and DOR receptors could be subjected to different degradation pathways. This hypothesis has been demonstrated for DOR that is well known to undergo endocytic trafficking to lysosomes [32,87]. However, conflicting data exist about the mechanism of agonist-induced opioid receptor down-regulation. Indeed, Chaturvedi et al. demonstrated that pretreatment with proteasome inhibitors, but not with lysosomal, attenuates the agonist-induced MOR and DOR down-regulation and that in the absence of agonist the proteasome inhibitors increase the steady-state levels of both opioid receptors [75] (Table 1). 

Hence, the exact mechanism by which chronic opioid agonists, including morphine, activate UPS machinery is still poorly understood, even though the involvement of UPS in neuropathic pain and morphine tolerance is envisioned [88,89]. In this regard, another aspect recently revealed is that the proteasome activity increase could be related to the phenomenon of analgesic tolerance. Indeed, it has been demonstrated that the co-administration of MG-132 with morphine prevents the development of morphine tolerance through the prevention of both spinal glutamate transporter down-regulation and spinal glutamate uptake activity decrease [90]. 

Furthermore, some studies have shown an increase of proteasome activity in neuropathic pain conditions. In this regard, it has been demonstrated that pain behavioral signs induced by spinal nerve ligation (SNL) are accompanied by the increase of dynorphin A levels in the spinal cord and that proteasome inhibitors are able to decrease painful signs together with dynorphin level normalization [89]. Moreover, authors have also demonstrated that proteasome inhibitors directly modulate the dynorphinergic system, since mouse insulinoma MIN6 cells exhibit a reduction in dynorphin secretion after epoxomicin and MG-132 exposure [89]. In addition, the single intrathecal injections of epoxomicin reduced capsaicin-evoked calcitonin gene-related peptide (CGRP) release from tissues of both sham-operated and SNL rats, thus, demonstrating the potential usefulness of proteasome inhibitors in the prevention and normalization of neurotransmitter release [89] (Table 1).

All these intriguing results converge on the involvement of UPS in the development and maintenance of neuropathic pain condition. Accordingly, we also obtained preliminary data in which we observed the activation of the proteasome degradation machinery in a neuropathic pain model induced by repeated exposure to oxaliplatin [91,92].

UPS involvement has also been proposed in the neurochemical effects of drugs of abuse [93,94], in neurodegeneration diseases [95,96], and, interestingly, also in addictive behaviors. Indeed, Massaly and coworkers demonstrated that the protein degradation process directed by UPS has a special role in a series of addictive behaviors, such as conditioned place preference, locomotor sensitization, and self-administration [97]. Authors have shown that morphine treatment evokes protein poly-ubiquitination in the synaptosomal fractions of the nucleus accumbens (NAc) and that mice subjected to the intra-NAc infusion of lactacystin or MG-132 did not show any preference for the morphine-associated compartment [97]. They also demonstrated that the intra-NAc injection of lactacystin obstructs the development of morphine behavioral sensitization, corroborating the hypothesis that behavioral sensitization may depend upon UPS activity. Moreover, lactacystin injection before each self-administration session induces animals to self-administer significantly less heroin compared to controls [97] (Table 1).

## 4. Modulation of Opioid Receptor Fate and Signaling

It is worth noting that opioid GPCRs internalization and their down-regulation are modulated by complex mechanisms. For instance, the 15-residue C-terminal deletion of DOR can block or slow the rate of receptor internalization process, but not its down-regulation [31,98,99]. Another point of interest is that opioid receptors, even though structurally homologous, can be differently regulated when activated by the same opioid agonist [100]. Moreover, different opioid receptors can be sorted by clathrin-mediated endocytosis following activation by the same agonist ligand [100]. In addition, the ability of different agonists to promote the internalization and desensitization processes of the same opioid receptor has been reported [16,101]. All these evidences clearly highlight the complexity of the signaling mediated by the activation of opioid receptors, that seems to be dependent not only on the opioid receptor type but also on the type of activating ligand.

In 1999, it was demonstrated that 100 nM [D-Ala2, D-Leu5]-Enkephalin (DADLE) exposure promotes DOR internalization after only 5 min, with a maximum intracellular accumulation after 20 min exposure. After more than 4 h, the intracellular DOR staining gradually diminishes, thus, suggesting the occurrence of a receptor degradation process [102]. In other words, acute agonist treatment may promote opioid receptor desensitization and endocytosis via phosphorylation [100,101,103], whereas the receptor fate following chronic exposure appears to be mainly associated with its degradation [102]. In this regard, further evidence highlighted the capability of opioid agonists to down-regulate the expression of genes codifying for opioid receptors [104,105], suggesting that cells reduce the synthesis of receptors involved in signaling following the continuous agonist stimulation. Thus, it seems that a cell’s response to continuous opioid agonist stimulation consists in reducing receptors biosynthesis and in their removal from the membrane first through the activation of internalization signals and after through degradation processes.

In this view, the contribution of ubiquitination process could be crucial in this complex regulation. It has been shown that opioid receptors are regulated by β-arrestin molecule recruitment which exists in two different forms: β-arrestin1 and β-arrestin2 (also known as arrestin2 and arrestin3, respectively) [17,20,106,107,108]. β-arrestins have been shown to act as scaffolds for both internalization and ubiquitination machinery phenomena [109,110].

In particular, it has been demonstrated that morphine is less effective in promoting MOR phosphorylation and β-arrestin2 recruitment compared to DAMGO and other opioids [107], indicating that the occupancy by different agonists promotes different receptor conformations. However, a GPCR kinase 2 (GRK2) overexpression might increase morphine-induced MOR phosphorylation, and, therefore, also β-arrestin2 recruitment and, eventually, MOR internalization [107]. Using confocal microscopy, Oakley and colleagues [111] showed that MOR may have a higher affinity for β-arrestin2 than β-arrestin1 when activated by highly potent opioids. 

With the aim of clarifying the different contribution of β-arrestin forms, further investigations showed that mice lacking β-arrestin2 exhibited numerous behavioral differences in response to morphine [112,113,114,115]. Among the altered behavioral responses, enhanced thermal antinociception, reduced antinociceptive tolerance, reduced constipation, reduced signs of withdrawal, enhanced dopamine release, and enhanced reward profiles were all observed. This evidence leads to the hypothesis that ligands that cause impaired β-arrestin2 recruitment could evoke potent analgesia with less severe side effects [107,116,117]. Based on this assumption, it has been speculated that the therapeutic and side effects elicited by morphine may depend by the recruitment of different β-arrestin forms, a phenomenon referred to as ‘biased agonism’ or ‘functional selectivity’ [118]. In turn, this different recruitment may exert a distinctive effect on the opioid receptor regulation itself [16,17,18,19]. However, other studies also demonstrated that only morphine’s effects are influenced by the lack of β-arrestin2, whereas other agonists maintain the same antinociceptive profiles in both wild-type and knock-out mice [112,113,114,115]. 

Other investigations focused their attention not only on the consequence of β-arrestin2 recruitment [119,120,121] but also on β-arrestin1, which appears critical for the MOR ubiquitination process since this ordinary post-translational modification does not occur in the absence of β-arrestin1 [122].

It is worth mentioning that DAMGO treatment promotes the recruitment of both β-arrestin1 and β-arrestin2 to MOR, and both arrestin forms are sufficient to promote, even alone, DAMGO-induced MOR internalization. In this situation, it seems that the DAMGO-induced internalization process takes place regardless of which β-arrestins are recruited, but it seems to rather depend on the ligand itself. Furthermore, DAMGO also promotes MOR ubiquitination which does not occur in the absence of β-arrestin1 [122], corroborating the hypothesis that the agonist-induced ubiquitination of GPCRs is dependent on β-arrestin1 form [123,124,125] and highlighting ubiquitination as a likely modification occurring after opioid agonists stimulation. In this situation, the β-arrestin1 recruitment necessary for receptor ubiquitination should serve as an adaptor between the MOR receptor, or other GPCRs, and E3 ligase enzymes.

In contrast, morphine recruits only β-arrestin2, thus, implying that this interaction could be sufficient to potentially produce MOR internalization but does not promote MOR ubiquitination, since morphine does not induce MOR–β-arrestin1 interaction [122]. The authors considered ubiquitination phenomena without clarifying MOR ubiquitination degree, thus, making it difficult to establish the receptor fate. Nevertheless, these results add extremely important information about the agonist-directed β-arrestin-mediated MOR regulation, although they do not yet clarify the precise role of proteasomal degradation in the modulation of signaling mediated by opioid receptor activation. Indeed, mono-ubiquitination is thought to mediate trafficking of receptors to the lysosome for degradation [109,126,127,128], whereas poly-ubiquitination may drive the substrate towards the 26S proteasomal degradation. However, it is relevant to point out that Groer and colleagues demonstrated that DAMGO, but not morphine, induces MOR ubiquitination, thus, suggesting that this post-translational modification is not an effect shared by all opioid agonists. 

Another point of interest is that the specific MOR / β-arrestin1 interaction has been shown to facilitate receptor dephosphorylation, which may represent an initial step in MOR resensitization process [122], thereby, contributing to the overall control of plasma membrane proteins and to the complex mechanism of opioid tolerance, as a consequence. In this picture, the possibility that opioid receptor poly-ubiquitination process could take place alternatively to mono-ubiquitination might help in understanding the observed effects evoked by preoteasome inhibitors, including the modulation of analgesia, tolerance [90] and addictive behavior [97]. 

## 5. Conclusions

Accumulating evidence indicates that a better understanding of opioid signaling modulation is relevant to control the adverse effects associated with the therapeutic use of opioid agonists. Encouraging results indicate the feasibility of proteasome inhibitors as adjuvants in different pathological conditions in which the role of the endogenous opioid system is relevant, such as drug abuse and pain. Moreover, the different ability of opioid agonists to recruit different arrestin forms, differently involved in the receptor ubiquitination process, suggest the potential role of UPS in the regulation of opioid receptor endocytic trafficking.

In this view, the contribution of UPS to the opioid receptors’ intracellular fate, as well as in their signaling, highlights the relevance of this degradation pathway. The observations here collected may offer knowledge for the development of new pharmacological tools and better pharmacological intervention strategies.

## Figures and Tables

**Figure 1 ijms-20-01441-f001:**
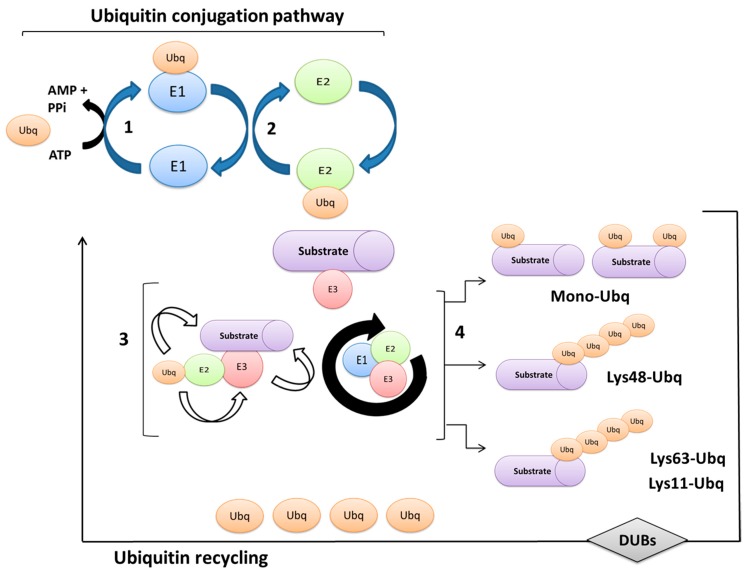
Schematic overview of the ubiquitin conjugation pathway. Ubiquitination is a three step process involving specific groups of enzymes, which are: (**1**) E1, ubiquitin (Ubq) activating enzyme; (**2**) E2, Ubq conjugating enzyme and (**3**) E3, Ubq ligase enzyme. Ubq is known to form covalent bonds with protein substrates (**4**) which, once modified, are subjected to different fates. De-ubiquitinating enzymes (DUBs) remove ubiquitins from substrate proteins.

**Figure 2 ijms-20-01441-f002:**
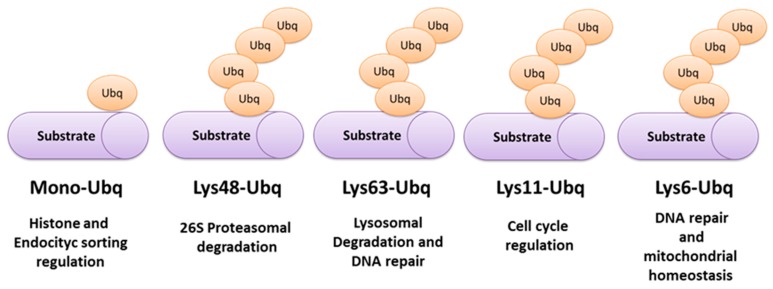
The ubiquitin post-translational modifications, either mono- or poly-ubiquitination, direct the conjugated substrates to different cellular fates which depend from the length and the type of ubiquitin chain.

**Figure 3 ijms-20-01441-f003:**
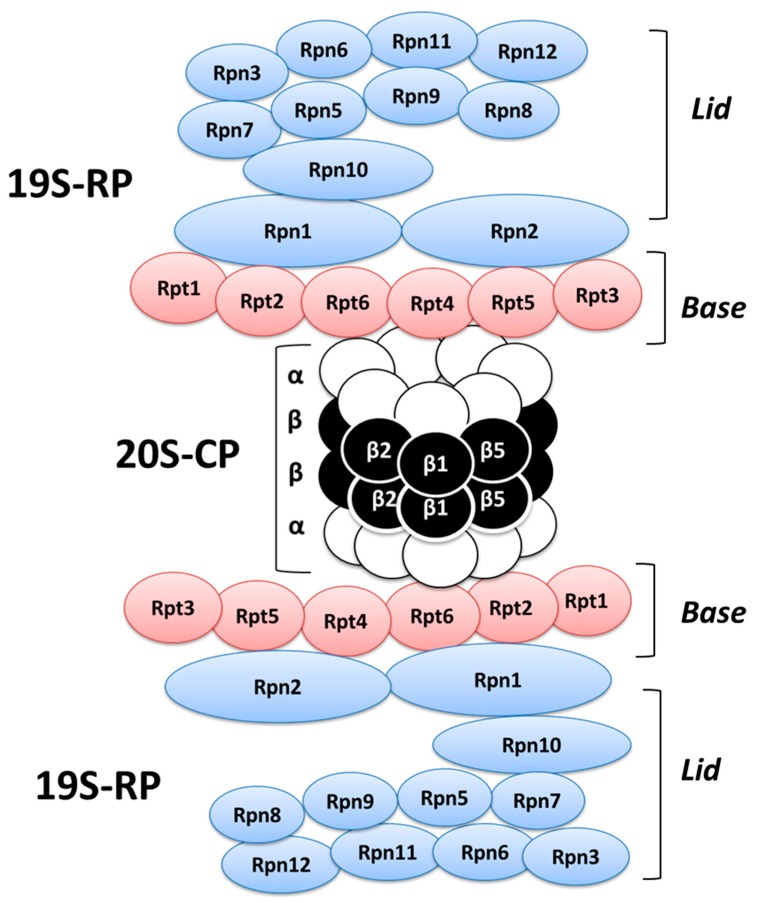
Schematic representation of the 26S Protesome complex.

**Table 1 ijms-20-01441-t001:** Proteasome involvement in analgesia, neuropathic pain, and addictive behavior.

Drugs and Treatments	Experimental Paradigm	Key Finding	References
ZLLL and lactacystin (proteasome inhibitors)	Human embryonic kidney 293 cells transfected with murine μ or δ receptors	Attenuation of the agonist-induced μ (MOR) and δ (DOR) down-regulation	Chaturvedi et al. 2001 [75]
MG-115 or lactacystin (proteasome inhibitors)	Human neuroblastoma SH-SY5Y cells	Suppression of Gβ down-regulation induced by prolonged morphine exposure	Moulédous et al. 2005 [76]
Opioid agonists	Human neuroblastoma SH-SY5Y cells	Significant increase of the proteasomal proteolytic activity	Caputi et al. 2017 [77]
MG-132 and lactacystin (proteasome inhibitors)	Human neuroblastoma SH-SY5Y cells	Block of the regulator of G protein signaling protein 4 (RGS4) reduction induced by DAMGO or DPDPE opioid agonists	Wang and Traynor, 2011 [86]
Co-administration of MG-132 with morphine	Adult male Sprague Dawley rats	Prevention of morphine tolerance development	Yang et al. 2008 [90]
Epoxomicin and MG-132 (proteasome inhibitors)	Neuropathic pain model (spinal nerve ligation)	Decrease of painful signs and dynorphin level normalization	Ossipov et al. 2007 [89]
Oxaliplatin exposure	Neuropathic pain model	Activation of the proteasome degradation machinery	Caputi et al. 2017 [91]
Lactacystin or MG-132 (proteasome inhibitors)	Opiate addictive behavior	Obstruction of morphine-associated compartment preference	Massaly et al. 2013 [97]
Lactacystin (protesome inhibitor)	Opiate addictive behavior	Obstruction of the morphine behavioral sensitization development and decrease of heroin self-administration	Massaly et al. 2013 [97]

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
