# Peer review of "Interplay between the Endogenous Opioid System and Proteasome Complex: Beyond Signaling"

_ijms, 2019, doi:10.3390/ijms20061441_

Round 1
Reviewer 1 Report
The manuscript deals with the role played by the ubiquitination processes and by UPS in the modulation of opioid receptor signaling and in pathological conditions involving the endogenous opioid system. In particular, the paper stresses the role of proteasome inhibitors in neuropathic pain, addictive behavior and analgesia. Hence, the manuscript nicely fits the topics covered by Int. J. of Molecular Sciences.
Before to merit publication, the manuscript needs minor corrections.
Introduction, it is stated that the endogenous opioid system comprises four major families of opioid ligands: β-endorphins, enkephalins, dynorphins and nociceptin/orphanin FQ. However, there is evidence that such peptides do not represent the endogenous ligands for MOR DOR KOR and NOP; indeed, some of these peptide classes do not show extraordinary selectivity, being rather promiscuous.
In the manuscript the Nociceptin receptor is basically presented as an opioid receptor, due to the high homology with the other opioid receptor; however this receptor stimulates pain transmission.
Figure 1 should be re-designed, in particular concerning lettering and small parts. Besides, some components of the figure are black on the black background, hence they are practically invisible.
For clarity, more figures are needed, for instance to better explain the structure and the particles of 26S proteasome, as well as a picture to support the efficacy of the paragraph 3
Conclusions, rephrase “a better understanding of opioid signaling modulation is relevant in order to avoid, or at least to control, the adverse effects ...” since “understanding” cannot “avoid”
Author Response
Ref: Ms. No. ijms-448990
Title: “Interplay between endogenous opioid system and proteasome complex: beyond signaling”
Reviewer #1
The manuscript deals with the role played by the ubiquitination processes and by UPS in the modulation of opioid receptor signaling and in pathological conditions involving the endogenous opioid system. In particular, the paper stresses the role of proteasome inhibitors in neuropathic pain, addictive behavior and analgesia. Hence, the manuscript nicely fits the topics covered by Int. J. of Molecular Sciences.
Before to merit publication, the manuscript needs minor corrections.
Q.1 Introduction, it is stated that the endogenous opioid system comprises four major families of opioid ligands: β-endorphins, enkephalins, dynorphins and nociceptin/orphanin FQ. However, there is evidence that such peptides do not represent the endogenous ligands for MOR DOR KOR and NOP; indeed, some of these peptide classes do not show extraordinary selectivity, being rather promiscuous.
A.1 A bunch of studies established that the opioid peptide receptors were heterogeneous and their pharmacological classification was confirmed when three mRNAs for the three receptor types were cloned and characterized (Evans et al., 1992 Science, 258: 1952-1955; Kieffer et al., 1992 Proc. Natl. Acad. Sci. U.S.A., 89:12048-12052; Chen et al., 1993 Mol. Pharmacol., 44: 8-12; Yasuda et al., 1993 Proc. Natl. Acad. Sci. U.S.A., 90:6736-6740). Subsequently, another opioid receptor was cloned, initially defined as ORL-1 (Mollereau et al., 1994 FEBS Lett 341:33-8) for which the endogenous ligand was isolated thereafter and named nociceptin (Meunier et al. 1995 Nature 377:532-5; Reinscheid et al., 1995 Science 270:792-794).
The β-endorphins, enkephalins and dynorphins differ in their affinity for μ, δ and κ receptors and have negligible affinity for NOP receptor (Zadina et al., 1997 Nature 386: 499-502). None of them exclusively binds one opioid receptor type thus in this perspective can be considered promiscuous; on the other hand, nociceptin binds NOP receptor only and do not show affinity for the classical opioid receptors (Meunier et al., 1995 Nature 377:532-5).
Nevertheless, the endogenous opioid peptides belonging to the four different families are currently and, by consensus, considered the natural ligands for the respective type of opioid receptors (Corder et al., 2018 Annu. Rev. Neurosci. 41:453-73).
Q.2 In the manuscript the Nociceptin receptor is basically presented as an opioid receptor, due to the high homology with the other opioid receptor; however this receptor stimulates pain transmission.
A.2 We understand the reasons generating this comment and, indeed, the role of nociceptin / NOP system in pain modulation has been quite long debated. After classical opioid receptor cloning, NOP was identified as a G-protein coupled receptor (GPCR) with 65% structure homology to the other members of the opioid family (Bunzow et al., 1994 FEBS Lett 347:284-8; Mollereau et al., 1994 FEBS Lett 341:33-8), thus defined as the fourth member of opioid G protein coupled receptor family.
NOP activation by nociceptin leads to Gαi coupling and signal transduction similar to that of opioid receptors. Surprisingly, initial studies attributed to nociceptin both pronociceptive and antinociceptive actions, depending on doses and routes of administration (Mogil and Pasternak, 2001 Pharmacol. Rev. 53:381–415; Mika et al., 2011 Neuropeptides 45:247-61). However, it has been rapidly understood that the apparent pronociceptive action of nociceptin peptide was basically due to its functional antagonism toward the neuronal pathways activated by classical opioid receptor stimulation (Mogil et al., Neurosci Lett. 1996 23;214:131-4; Mogil et al., Neuroscience. 1996 75:333-7).
For example, intracerebroventricular (i.c.v.) administration of nociceptin exerts an anti-opioid action (Schulz et al., 1996 NeuroReport 7:3021-3025), this pronocicective action is not present in NOP knock-out mice and it is hampered by using specific NOP receptor antisense oligonucleotides (Calò et al., 2000 Br. J. Pharmacol. 129:1261-1283). In contrast with the effects obtained following i.c.v. injection, when administered intrathecally (i.t.) at the spinal level, nociceptin has an antinociceptive effect similar to that given by a classical opioid agonist, including weakening and sedation (Xu et al., 2000 Peptides 21:1031-1036). Indeed, an analgesic effect was observed in the rat following intrathecal administration of PheΨ, a partial NOP receptor agonist and nociceptin analogue (Candeletti et al., 2000 Life Sci. 66:257-64). This same dualism also concerns its involvement in drug addiction. Indeed, NOP receptor agonists have promising efficacy for attenuating the rewarding effects of morphine and cocaine (Zaveri et al., 2018 Front Psychiatry. 2018 Nov 29;9:638), even though it has been also shown that NOP antagonist may reduce ethanol self-administration and ethanol seeking in animal models (Rorick-Kehn et al., 2016; Alcohol Clin Exp Res. 40:945-54).
Although the mechanisms mediating the different actions of nociceptin are not yet completely clear, this endogenous peptide is considered a fully-fledged component of the endogenous opioid system.
Q.3 Figure 1 should be re-designed, in particular concerning lettering and small parts. Besides, some components of the figure are black on the black background, hence they are practically invisible.
A.3 We apologize for the problem related to the Figure 1 where, as correctly pointed out by the Reviewer, some components appear as black on black background. Hence, we modified the file format. In addition, since some components in the figure appeared too small, we reorganized the Figure 1 making it more readable (see page 3 in the revised version of the Manuscript, paragraph 2 “Ubiquitination processes and ubiquitin-proteasome system (UPS)”, Figure 1). We have also added a new Figure 2 (using a part removed from Figure 1) to provide a sequence of figures that could help the reading of the paragraph (see page 4 in the revised version of the Manuscript, paragraph 2 “Ubiquitination processes and ubiquitin-proteasome system (UPS)”, Figure 2). Finally, we have also elucidated in the text some aspects of the ubiquitination and proteasome-mediated degradation process that were not sufficiently described (see page 2, lines 76-82; page 3, lines 108-110).
Q.4 For clarity, more figures are needed, for instance to better explain the structure and the particles of 26S proteasome, as well as a picture to support the efficacy of the paragraph 3.
A.4 According with the Reviewer suggestion, we reorganized the figures to better guide the readers throughout the Manuscript. Thus, we added a new Figure 3 where the functions of different subunits that form the 26S proteasome complex are elucidated (see page 5 in the revised version of the Manuscript, paragraph 2 “Ubiquitination processes and ubiquitin-proteasome system (UPS)”, Figure 3). This aspect has also been clarified in the text (see page 4, lines 128-146).
Moreover, we agree with the Reviewer that the paragraph 3 needs some support for a better readability. For this reason, we added the Table 1 gathering experimental evidences, and related references, to provide a well-defined picture of the evidence available to date (see page 7 in the revised version of the Manuscript, paragraph 3 “Involvement of UPS machinery in the opioid receptor signaling associated with analgesia, neuropathic pain and addictive behavior”, Table 1).
Q.5 Conclusions, rephrase “a better understanding of opioid signaling modulation is relevant in order to avoid, or at least to control, the adverse effects ...” since “understanding” cannot “avoid”
A.5 We agree with the Reviewer that the terms “understanding” and “avoid” in the same sentence could be misleading for the readers. In this view, we modified the sentence accordingly (see pag 9 in the revised version of the Manuscript, paragraph 5 “Conclusions”).
Reviewer 2 Report
The authors seem to discuss the role of UPS machinery in the opioid signaling. They argue the role of ubiquitination on opioid receptors. Their discussion is a little confusing. It would be better to discriminate ubiquitination for internalization of cell surface receptors targeted to lysosomal degradation from ubiquitination for proteasomal degradation. Proteasomal degradation of signaling molecules related to cell surface receptors may affect the function of receptors. However, it is unclear whether proteasome complex may involve directly the function of opioid receptors after those receptors express on the cell surface. It is also unclear whether proteasomal complex may involve directly the degradation of opioid receptors after those receptors express on the cell surface. Proteasomal degradation is a ubiquitous mechanism in all proteins within a cell and in all types of cells in a body. Inhibitors for proteasomal degradation may have various functions including severe adverse effects on a body.
Author Response
Reviewer #2
Q.1 The authors seem to discuss the role of UPS machinery in the opioid signaling. They argue the role of ubiquitination on opioid receptors. Their discussion is a little confusing. It would be better to discriminate ubiquitination for internalization of cell surface receptors targeted to lysosomal degradation from ubiquitination for proteasomal degradation. Proteasomal degradation of signaling molecules related to cell surface receptors may affect the function of receptors. However, it is unclear whether proteasome complex may involve directly the function of opioid receptors after those receptors express on the cell surface. It is also unclear whether proteasomal complex may involve directly the degradation of opioid receptors after those receptors express on the cell surface. Proteasomal degradation is a ubiquitous mechanism in all proteins within a cell and in all types of cells in a body. Inhibitors for proteasomal degradation may have various functions including severe adverse effects on a body.
A.1 We understand the reasons generating these comments that are also due to the few information still available about the opioid receptor ubiquitination process. Nevertheless, the cited experimental evidence points to an involvement of proteasome activity in different opioid-mediated responses. We agree with the Reviewer that clarifying the role of ubiquitination in internalization and degradation processes is fundamental; however currently available evidence do not allow drawing a satisfactorily clear picture. Indeed, as we reported in the Manuscript, different authors do not specify the degree of ubiquitination evaluated in their experiments and this fact makes it difficult to clarify the role of proteasomal degradation in the modulation of signaling mediated by opioid receptors activation. We suggest that the effects evoked by preoteasome inhibitors, including the reduction of the development of analgesic tolerance and the modulation of addictive behavior, could involve the proteasomal degradation process. Certainly, we are aware that this kind of pharmacological approach is not free of adverse effects; however a better understanding of the interplay between endogenous opioid system and proteasome complex could prove to be useful in the control of these pathological conditions.
In the revised version of the text, we made some changes to improve the readability of the Manuscript.
Round 2
Reviewer 2 Report
The revised manuscript is ready to accept.